# Using authentic representations of practice in teacher education: Do direct instructional and problem-based approaches really produce different effects?

**Jürgen Schneider**[1]*, **Marc Kleinknecht**[2], **Thorsten Bohl**[1], **Sebastian Kuntze**[3], **Markus Rehm**[4], **Marcus Syring**[1]

1 Department of Education, University of Tübingen, Tübingen, Germany, 2 Department of Educational Sciences, Leuphana University Lüneburg, Lüneburg, Germany, 3 Department of Mathematics and Informatics, University of Education Ludwigsburg, Ludwigsburg, Germany, 4 Department of Chemistry, School of Education Heidelberg, Heidelberg, Germany

* juergen.schneider@uni-tuebingen.de

**Data Availability Statement:** https://doi.org/10.4232/1.13468.

## Abstract

This paper investigates the effects of different instructional approaches (problem-based vs. direct instructional) on student teachers' analysis of practice when using authentic representations of practice in teacher education. We assigned 638 student teachers from 21 equivalent teacher education courses to one of the two conditions. Students' analyses of practice were evaluated on selective attention, reflective thought, and theory-practice integrations in a pre-post-design. In line with inconsistent findings from prior research, we were able to produce evidence for equivalent effects of the instructional approaches on all dependent variables using Bayesian data analyses. As called for in a review on the topic, we additionally explored the role of the instructors administering the field study interventions. Findings revealed that a positive attitude toward the instructional approach the instructors administered was related to more theory-practice integrations in the students' analyses.

## 1. Introduction

Learning from practice is a key element of professionalization in teacher education [1]. Over the past 15 years, approaches that enable student teachers to learn from practice by using representations of practice (particularly via video) have steadily increased. Representations of practice can be described as a window into practice that enables students to experience and understand teaching [2, 3]. They are considered to enable students to approximate practice in a controlled environment and thus prepare them for professional action [4]. Representations of practice can only realize their potential for professional development if they are paired with a substantive reflection that is not limited to surface features [5–8]. However, it is challenging for student teachers to engage in deep reflection without further support [9]. Novices tend to focus overly on the surface features of classroom interactions [10] and themselves rather than on student learning [11]. To address this challenge, researchers aim to identify key processes

**Funding:** This work was supported by the Ministry of Science, Research and Arts of the state Baden-Württemberg, Germany The funders had no role in study design, data collection and analysis, decision to publish, or preparation of the manuscript.

**Competing interests:** The authors have declared that no competing interests exist.

of reflection to make it tangible in teacher education contexts [12]. Based on seminal literature on reflection by Dewey [13], research in teacher education programs typically emphasizes processes of selective attention, reflective thought, and theory-practice integration. A variety of programs have been able to demonstrate that these skills of selective attention, reflective thought, and theory-practice integration can be fostered through interventions in teacher education [14, 15]. An open question concerns the comparison of different intervention programs and their effectiveness, answering the question: Under which boundary conditions are interventions effectively promoting reflective engagement with representations of practice? In the current study, we compare different instructional approaches and their effects on selective attention, reflective thought, and theory-practice integration.

## 1.1 Selective attention, reflective thought, and theory-practice integration with authentic representations of practice

*Selective attention*, as defined by Sherin and van Es [6], describes the ability to perceive and be aware of relevant, in-depth features of instruction. What is considered *relevant* depends on the curricula of the teacher education programs and thus on the respective definition of teacher professionalism. In our case, we address classroom management, which is a broadly recognized dimension of teaching quality [16].

The rationale behind the need to train selective attention is that teachers cannot consciously reflect on (or respond to) relevant aspects of instruction if they do not notice them [17]. Comparisons between experienced teachers and student teachers reveal that student teachers tend to have less selective attention to relevant instructional features [18, 19]. At the same time, teaching experience is not a sufficient condition for the development of selective attention to relevant features of instruction [20]. Even experienced in-service teachers may attend to surface features when viewing representations of practice if they have not been trained in selective attention [21]. However, robust evidence exists that selective attention can be trained in teachers [22] and student teachers [23] through the use of representations of practice. In their syntheses of research, Marsh and Mitchell [24], as well as Gaudin and Chaliès [14] found that the use of video-based representations of practice has the *potential* to promote selective attention among pre-service and in-service teachers.

Selective attention creates the possibility for subsequent *reflective thought*. From a theoretical stance on reflection, reasoning about different possible options for the selected situation constitutes reflective thought [6, 13, 25]). In the context of learning from representations of practice, reflective thinking involves describing a situation, exploring options for the situation, anticipating consequences, and making a decision based on these processes. These processes constitute the core of reflection in the teaching profession. Learning to apply these processes to practice represents a necessary prerequisite for professional development in the teaching profession [26]. Based on their research synthesis, Marsh and Mitchell [24] argue that collaborative learning with video-based representations of practice is particularly effective in promoting reflective thought due to its discursive nature that scaffolds the learning process. Likewise, Gaudin and Chaliès [14] identified a series of empirical studies that corroborate the effectiveness of video-based learning arrangements in this regard.

Topic-specific selective attention and subsequent reflective thinking on representations of practice in teacher education create opportunities to *integrate theory and practice*. Integrating theory and practice is pivotal to the professional development of teachers [27]. On the one hand, it helps to avoid a purely theory-based education that leads to practice shock [28]; on the other hand, it prevents over-simplified adoptions of practice routines that are disconnected from scientific knowledge about teaching [8]. Representations of practice provide a possibility

to foster relations of theory and practice through contextualization and abstraction during the process of reflective thinking [3, 29]).

## 1.2 Problem-based and direct instruction

The instructional approach in which authentic representations are leveraged plays a pivotal role in the effectiveness of the learning outcome [30]. These instructional approaches may have differential effects on how learners perceive and reason about practice [31]. Even though most learning arrangements using authentic representations in teacher education follow a problem-based (PB) approach, it is still unclear which differential effects direct instruction (DI) will evoke, due to the lack of empirical studies that systematically compare these two approaches.

The use of authentic representations is an inherent part of a PB approach. It opens a problem space that initiates the learning process [32]. Learners explore the problem and apply problem-solving strategies, such as hypothesizing and self-regulated research and application of knowledge [33]. PB approaches involve small groups, whose members divide tasks, share knowledge and engage in discursive reasoning to reach a conclusion [34]. Since reasoning constitutes an essential part of PB learning, this instructional approach has the potential to promote learners' reflective thought through practice [35]. In contrast, DI describes a teacher-centered approach in which phases of modeling are typically followed by phases of guided and individual practice [36]. Within DI, authentic representations are used to exemplify a solution presented previously by the instructor [37]. Learners may subsequently work on similar authentic representations to replicate the solution process, and thus, do not necessarily reason about different solutions. Thus, from a theoretical perspective, DI would be suitable to practice the selective attention of participants. Moreover, DI as a structured approach may be particularly helpful for novice learners, who may suffer from cognitive overload otherwise [38].

## 1.3 Empirical findings on effects of different instructional approaches

Research over the past several decades on different instructional approaches suggests that DI is successful in promoting content knowledge and transfer [36], particularly in novices [39]. On the other hand, there is evidence that PB instruction promotes skill rather than knowledge acquisition [40] and may foster other factors relevant to the learning process, such as motivation or attitudes [41]. In our study, we compare these two instructional approaches and their effects on student teachers' selective attention, reflective thought, and integration of theoretical knowledge with representations of practice. To our knowledge, there are only two studies directly comparing different instructional approaches with respect to these variables.

Seidel, Blomberg, and Renkl [42] compared two different instructional approaches that share characteristics with DI and PB—rule-example versus example-rule. In the rule-example strategy, definitions of the topic (goal clarity, scaffolding, and learning climate) were given and subsequently exemplified in a classroom video. The students were then asked to practice the demonstrated observation on further video representations. The example-rule strategy used group discussions stimulated by observations on classroom videos. These discussions and a subsequent moderated collection of ideas were intended to lead to an understanding of the topic. Aggregated scores on noticing and evaluating classroom situations showed that the rule-example group outperformed the example-rule group. Unfortunately, the paper did not report separate scores for noticing and evaluation; thus, it is challenging to draw more precise inferences from the effects. In a test on factual knowledge, the rule-example group outperformed the example-rule group. However, when asked to plan a lesson, the example-rule group more frequently elaborated on theoretical ideas that included situational references rather than being merely general.

Barth et al. [25] compared two collaborative instructional approaches using authentic representations of practice (vignettes) that included either direct instructional or traditional problem-based elements. Interventions were implemented in professional development courses for student teachers on the topic of classroom management. In the DI condition, students received lecture-like instruction on theoretical aspects of classroom management, then analyzed several vignettes guided by a three-step worksheet that depicted selective attention, reflective thought, and theory-practice integrations. In the PB condition students first observed one problematic vignette, then independently read theoretical literature on classroom management in a self-study phase. Afterward, they proceeded to analyze the given vignette using the same worksheet as the DI condition. The intervention, therefore, focused more on different approaches concerning the acquisition of theoretical knowledge and less on different approaches concerning the analysis process. Measures included selective attention and theory-practice integrations in written analyses on video-based classroom vignettes. Posttest comparison of the groups in the first study yielded no differences in measures of selective attention and a small to medium effect on theory-practice integration favoring the DI group. In a second, study Barth et al. [25] focused solely on the DI group revealing no differences in selective attention and a strong effect on theory-practice integrations from pretest to posttest.

## 1.4 Empirical findings on the effects of instructors' attitudes

In a systematic review, Baecher et al. [43] highlighted the role of instructors administering the interventions in field studies, which is rarely clarified in scholarly publications. Guiding discussions and structuring analyses of participants, the instructor plays an important part in the success of the learning process [44, 45].

One important aspect when implementing and studying the effects of teaching approaches in field studies is the attitude of instructors administering the treatment conditions. Instructors of teacher education courses may differ in their attitudes about teaching styles and learning scenarios [46]. These attitudes can be consistent or inconsistent with features of the treatment conditions and influence the way instructors practice their teaching. Even when facing obstacles (e.g., being told to teach differently), instructors may try to maintain consistency between their attitudes and their practice [47]. Given that field studies involve instructors in the delivery of treatments, their attitudes toward the treatment can be an important source of information and predict outcomes. Several studies revealed that a positive attitude of instructors toward the treatment may have an effect on the subject's performance concerning critical measurements (e.g., seminal first findings by Rosenthal and Fode [48]). Since there has been little research in this regard in teacher education field studies, this part of our study is exploratory.

Based on the findings described above, we conducted an experimental field study comparing different instructional approaches and their effects on selective attention, reflective thought, and integration of theoretical knowledge with authentic representations of practice. We also address the role of the instructor, focusing on their attitude toward the treatment.

There exists a large body of exploratory research on learning arrangements utilizing video-based representations of practice in teacher education. However, to accumulate evidence on differential effects and border conditions of video-based approaches, we need studies conducting experimental variation and comparison.

## 1.5 Research questions and hypotheses

In this study, we investigate how different instructional approaches (PB vs. DI) and the instructors' attitudes are related to student teachers' selective attention, reflective thought, and integration of theoretical knowledge with authentic representations of practice. One of the

hypotheses on reflective thinking (H2$_1$) was based on strong assumptions derived from theory. The rest of the hypotheses were labelled as exploratory, since robust research is lacking.

*RQ1 on selective attention*: To what extent are different instructional approaches (DI and PB) and instructors' attitudes on these approaches related to the selective attention of student teachers?

Based on theoretical considerations and first empirical findings we hypothesize that DI and PB instructional approaches both foster selective attention (no difference between groups), with instructors' positive attitudes about the instructional approach positively related to selective attention (H1$_1$: $\beta_{treat} = 0$ & $\beta_{IA} > 0$).

*RQ2 on reflective thought*: To what extent are different instructional approaches (DI and PB) and instructors' attitudes on these approaches related to the reflective thought of student teachers?

Based on established research findings, we hypothesize that a PB instructional approach leads to more elaborate reflective thought compared with DI, with instructors' positive attitudes about the instructional approach positively related to reflective thought (H2$_1$: $\beta_{treat} > 0$ & $\beta_{IA} > 0$).

*RQ3 on theory-practice integrations*: To what extent are different instructional approaches (DI and PB) and instructors' attitudes on these approaches related to theory-practice integrations of student teachers?

We hypothesize that DI and PB instructional approaches both lead to similar amounts of theory-practice integrations (no difference between groups), with instructors' positive attitudes about the instructional approach positively related to theory-practice integrations (H3$_1$: $\beta_{treat} = 0$ & $\beta_{IA} > 0$).

Consistent with our hypotheses, we use Bayesian inferential statistics in our data analysis. This allows us (in contrast to frequentist null hypothesis testing) to make statements about the equivalence of groups as used in H1$_1$ and H3$_1$ [49]. We tested the hypotheses of the two predictors (instructional approaches, instructors' attitudes) within each dependent variable simultaneously to increase rigor by making the predictions as precise as possible.

## 2. Method

Written approval has been obtained from the Ethics Committee of the Faculty of Economics and Social Sciences at the University of Tübingen (without approval number). The participants of the study were informed about the study's research interest one month before the start of the study. Participants completed a written informed consent form which included a privacy statement. Participants were aware that participation was voluntary and non-participation had no consequences. We also informed participants that we will anonymize the dataset after data collection and will not share it with the instructors.

### 2.1 Participants

In total, 638 student teachers participated in the study, recruited from 21 introductory courses for secondary teachers. The 21 courses covered the same content on teaching, learning, and instruction over one semester (15 weekly 90-minute sessions). They all took place in the same academic semester and teacher education program, allowing for comparable conditions. The treatment was a regular part of the course, measurements were administered through an online pretest and posttest survey. Participants were $M_{age} = 21.01$ ($SD_{age} = 2.36$) years old on average and predominantly female (65.0% female, 33.2% male, and 1.8% "other" or none). Students usually attend this course in their second of 10 semesters; hence, 72.3% of the participants were studying in their second semester.

## 2.2 Design

The study was conducted in a regular teacher education program in Germany, allowing for field study conditions. In each of the 21 courses included in the study, one of the two interventions was implemented. For the interventions, we redesigned part of the courses (two sessions and an inter-session assignment) using authentic representations of practice. For this purpose, we focused on sessions in which classroom management was on the curriculum. These were sessions six and seven of 15 sessions. The other 13 sessions before and after the treatment remained as planned by the instructors ("business as usual"). Courses were either realized as problem-based ($n_{PB}$ = 11) or direct instructional ($n_{DI}$ = 10). Thus, the interventions were implemented at the course level. Students were assigned to the courses via a university-administrated system we had no influence on. We allocated courses to the different interventions such that each intervention was uniformly distributed across days between Monday and Friday, time slots (8am to 8pm) and the six instructors. Each instructor taught DI an PB courses; the conditions were balanced within the instructors (teaching the same amount of both conditions, except when teaching an odd number of courses). These instructors were the same as those teaching throughout the semester, further strengthening field study conditions. We used the R package 'BayesFactor' [50] to test whether conditions were equivalent concerning several potentially confounding context variables. The Bayes factor is a measure of relative evidence comparing two hypotheses, one of which can be specified as a null hypothesis. As opposed to (classical frequentist) null hypothesis significance testing, Bayes factors allow us to gather relative evidence for a null hypothesis, and therefore, test for equivalence. In our Bayes factor analyses for two independent samples with default priors of the BayesFactor package [50], we tested the conditions for equivalence ($H_0$) or differences ($H_1$) of the groups concerning gender ($BF_{10}$ = .109), teaching experience ($BF_{10}$ = .116), experience with video-based analyses ($BF_{10}$ = .099), topic-related literature read ($BF_{10}$ = .114), and prior knowledge on classroom management theories ($BF_{10}$ = .101). All Bayes factors pointed toward evidence of equivalence between the groups prior to the treatment.

## 2.3 Treatments, materials, and procedure

The procedure of the treatment is shown in Table 1. To ensure that the instructors were well trained in conducting the intervention sessions, they taught these sessions one semester before the study and received feedback from two researchers videotaping and observing the sessions. In multiple subsequent meetings with all instructors, we were able to further optimize treatment compliance. Choosing from a pool of 14 normal-practice classroom videos, we found four vignettes suitable as authentic representations of classroom management, each with a duration of approximately 5 minutes. These vignettes were deemed appropriate because strategies of classroom management were particularly visible in them (thus providing an authentic representation) and were appropriate for the student's grade level (secondary education). To offer text-based vignettes, we transcribed the video vignettes and added nonverbal information. Text and video vignettes were evenly distributed between the two treatment groups, representing another factor (2 x 2 design) not considered as a predictor in this paper (for description and results see [51]). One vignette was used in the first session, another as an assignment (to be completed until the second session), and two in the second session.

The first session had two parts. In the first part, the instructor introduced students to the topic of classroom management and its theoretical approaches [52–54]. Instructors gave definitions and clarified strategies concerning classroom management in a short exercise. Subsequently, students were made familiar with the steps of how to analyze the authentic representations of practice (i.e. vignettes): "Describe the problem/situation, describe the

**Table 1. Procedure for measurement and intervention.**

| | *Problem-based learning (PB)* | *Direct instruction (DI)* |
|---|---|---|
| *Before intervention* (40 minutes) | (Online) pretest (analysis of practice, demographics, and covariates) | |
| *Session 1* (90 minutes) | Instructor lectures on classroom management theories and steps for analyzing classroom vignettes | |
| | • Students analyze video or text-based vignette in small groups, focusing on classroom management<br>• Students participate in collaborative discussions about interesting situations | • Instructor analyzes a video or text-based vignette on classroom management, step by step<br>• Students replicate the analysis with a new situation |
| *Assignment* (60 minutes) | Analysis of a video or text-based vignette at home | |
| *Session 2* (90 minutes) | • Students analyze video or text-based vignette in small groups, focusing on classroom management<br>• Students participate in collaborative discussion about interesting situations | • Instructor analyzes a video or text-based vignette on classroom management, step by step<br>• Students replicate the analysis with a new situation |
| *After intervention* (40 minutes) | (Online) posttest (analysis of practice, covariates) | |

teacher's action, reason about alternative courses of action, anticipate reasons and consequences of these actions, decide on one of the alternatives". This first part was taught equally in all courses and in both conditions. In contrast, the second part varied between the two conditions.

**2.3.1 Direct instructional treatment.** In the second part, instructors introduced students to the lesson represented in the vignette, giving background information on the topic, the lesson structure, the class level, and the specific sequence of the lesson they were about to witness. The vignette was then shown to the entire course without interruption so that students could visualize the classroom activities. In a second round, the instructor stopped at certain situations and demonstrated a step-by-step analysis that integrated theoretical aspects of classroom management. After the situation was sufficiently analyzed, the instructor continued by repeating the analysis with several more situations. After this, students individually analyzed some more situations chosen by the instructor. These analyses were performed as a dialogue with the instructor and other students contributing. In the second session, the same procedure of demonstration and exercise was repeated for two additional vignettes.

**2.3.2 Problem-based treatment.** Problem-based courses also started with the instructor giving background information on the subsequent vignette. After that, the vignettes were handed out either as text vignettes on paper or video vignettes viewed on laptops. Students came together in groups of four to five members. They observed several situations and discussed the analysis in their group. The students, not the instructor, selected which situations they analyzed. Students were free to determine what they consider to be a problem (selection of situations) and what was and was not part of the problem. Students were also free in inquiring about these situations. To guide the analysis, students received key questions that targeted the analysis of practice steps. These key questions merely served as a guide in case students needed support with their analysis. The questions did not serve as step-by-step instructions and were not introduced as such. In a final discussion, the whole course talked about two or three of these situations. Students were asked to describe situations that stood out to them and perform analyses of them. The instructor acted as a moderator and tried to include different student suggestions about the situation and promote a discursive discussion. In the second session, the instructor repeated the same procedure of small group discussion and final course group discussion with two more vignettes.

Treatments checks were administered by one of three trained raters who visited the treatment sessions. They judged the implementation of the treatment on eight items that measured the degree of DI of the instructor on a 6-point Likert scale (from "doesn't apply at all" to "applies completely"). A (reversed) example item is as follows: "Students chose which situations to have a closer look at while working with the vignettes." Raters also recorded the time students were effectively able to work on the vignettes. For both measures, the inter-rater reliability showed good intraclass correlation (ICC = .96–.99). Internal consistency concerning the treatment check scale showed good scores (Cronbach's α = .96 for both sessions). In both sessions, we found evidence of equivalence between the groups for the time students worked on the vignettes (1$^{st}$ session: $BF_{10}$ = .514; 2$^{nd}$ session: $BF_{10}$ = .545). Evidence in both sessions is not very strong, however, in both cases they point in the same direction. In contrast, we were able to provide evidence of differences in the degree of direct instruction for the treatments (1$^{st}$ session: $BF_{10} = 3.76 \cdot 10^6$; 2$^{nd}$ session: $BF_{10} = 2.34 \cdot 10^5$) using Bayes factor analyses for two independent samples with default priors of the BayesFactor package [50]. These findings corroborate that the instructors realized the different instructional methods as planned with comparable times on task.

## 2.4 Measures

Dependent variables and covariates were assessed via an online survey the students were asked to complete as part of the course, before (pretest) and after (posttest) the two treatment sessions. As stated in the research questions, we are looking into the dependent variables of selective attention, reflective thought, and integration of theoretical knowledge with authentic representations of practice. These three variables were assessed through written comments that students gave on classroom vignettes presented in the survey. Students were instructed to comment on situations they perceived relevant in the topic of classroom management. Since students might have been familiar with the idea but not with the term classroom management in the pretest, we asked them to observe the lesson planning, control of behavior, and shaping of relationships witnessed in the vignettes—the three dimensions of classroom management that were taught in the treatment sessions. They were instructed to write and save each analysis separately using a "save comment" button below the text box (Fig 1). After saving an analysis, they were free to continue the observation and write further analyses.

The pretest and posttest each presented three vignettes of a classroom video that added up to ten minutes. To avoid a test effect, different vignettes from the same videotaped classroom lesson were used in the pretest and posttest. We investigated the extent to which pairs of

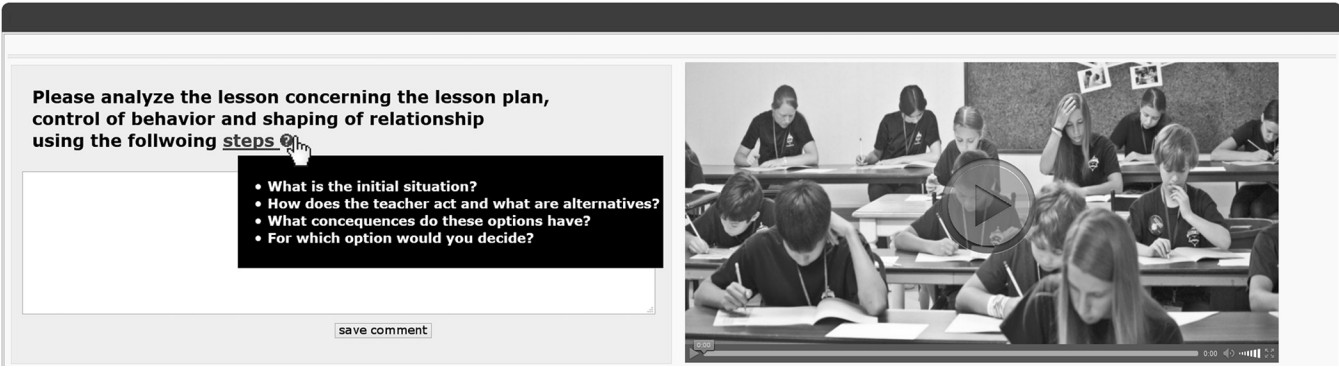

**Fig 1. Screenshot of the web-based survey: One of three vignettes to be analyzed by students.**

similar vignettes could be found from a classroom video, each of which was then split between the pretest and the posttest. Three experts (in the fields of practice-oriented teacher education and classroom management) rated the vignettes on 14 dimensions (e.g., complex, interesting, or classroom management) with a mean (standard deviation) agreement of $r_{WG}$ = .79 (.31). We used these ratings to conduct cluster and graphical analyses to find three pairs of similar vignettes that were then separately allocated to either the pretest or posttest.

**2.4.1 Number of selected situations as selective attention.** In teacher education programs, what is defined as relevant may depend on the current learning goal of the course or learning arrangement (e.g., teacher guidance and support [55]). In our study, we focus on *classroom management*; the topic of the two sessions in the course our treatment took place.

Thus, we operationalize selective attention as situations the participants select to discuss a relevant topic, where "relevant" means that the selected situations are instances of the focused topic of our study, "classroom management." We use the number of selected situations in the web-based survey students commented on (i.e., number of saved analyses). Participants were instructed to comment on every situation they perceived as being relevant concerning the classroom management in the vignette. Therefore, they were able to save as many analyses as they pleased. A trained coder rated whether the analyses focused on classroom management or were off-topic. Counts were summed up for each of the three vignettes in the pretest and posttest, thus constituting the number of selected (and analyzed) situations for each vignette.

**2.4.2 Realized inquiry steps representing reflective thought.** We operationalized reflective thought by students' ability to apply the inquiry steps to the selected situations from the vignettes in the online survey. On the survey, students were reminded about the inquiry steps in the item question (see Fig 1). Thus, we measured whether students could *transfer and apply* these steps to the practice situations observed rather than whether they remembered the steps. The comments written and saved by the students were coded as to whether they contained the individual inquiry steps (dichotomous). The scale from 0 to 3 categorizes whether we detected none, one, two, or all of these steps in each of the student's written comments (Table 2).

Inter-rater reliabilities for all codings were computed based on randomly selected 20% of the approximately 7 600 comments written by the participants in the pretest and posttest.

**Table 2. Levels of reflective thought based on coded inquiry steps.**

| Level | Steps realized | Example |
|---|---|---|
| 3 | • Description, alternatives, and consequences | The teacher collects the notes from the students. It becomes increasingly loud in the classroom as the teacher hangs notes on the blackboard. The teacher gives the students a warning. It gets quieter. Overall, it would be better to include the students more in hanging up the notes. That would enable a quieter learning situation. |
| 2 | • Description and alternatives | Students insult each other, but the teacher tries to ignore it by continuing with the lesson. She should really try to stop the insults before recommencing. |
| | • Description and consequences | While checking the results of the assignment, the teacher praises the students a lot. That'll motivate them. |
| | • Alternatives and consequences | The class is loud as some students begin to present their results. The teacher should not let them begin to present until the rest of the class is quiet. Thus, half the information wouldn't be lost. |
| 1 | • Description | There was quite a bit of noise in the classroom, then the teacher placed her finger on the lips. |
| | • Alternatives | The teacher should react to the student wandering around the room. |
| | • Consequences | The students don't seem to heed the teacher's actions. |
| 0 | none | Everything seems normal. |

Cohen's Kappa scores of the two trained raters were satisfactory (κ = .64–.77) and disagreements were resolved through discussion. The raters' scores were tested for one-dimensionality per vignette using confirmatory factor analysis (CFA) with individual comments used as indicators and the robust maximum likelihood estimator (MLR) obtained by full information maximum likelihood [56]. The data showed a good fit regarding all six vignettes, $p[\chi^2]$ = .13–.99, CFI = .95–1, RMSEA ≤ .001–.03, $p$[RMSEA < .05] = .86–1. Furthermore, model comparisons indicated that we were able to assume strict measurement invariance for all vignettes between treatment groups. In addition, reliability between the comments revealed good internal consistency for all vignettes, McDonald's ω = .70–.80 [57]. Thus, we computed mean scores for each vignette in the pretest and posttest, reflecting the average quality of reflective thought.

**2.4.3 Theory–practice integration.**   We assessed students' theory–practice integration regarding classroom management principles by coding their written analyses for terms and principles from the classroom management literature. When detected, coders evaluated whether the theoretical principle fit the situation to which the comment referred. We did not decide to use a sample solution (a predetermined list of theoretical principles associated with specific situations) because it would not do justice to the complexity of classroom situations. There can be no certainty about what students were truly referring to within these situations when only coding for predetermined theoretical principles while excluding further information from the participants' view. Written analyses that referred to theoretical principles of classroom management that fit the situation described were coded dichotomously with 1 if they met these criteria (e.g., "the teacher does not manage to show withitness by maintaining eye contact and keeping students under control or quiet") or 0 if they did not fulfill these criteria.

Inter-rater-reliability was satisfactory (κ = .74). We computed mean scores for each vignette in the pretest and posttest, reflecting the percentage of comments that included theoretical principles of classroom management.

**2.4.4 Theoretical literature on classroom management.**   We allege that the students were unfamiliar with the topic of classroom management since the teacher education program's curriculum did not cover it up to this point (second semester). Concerning the two sessions on classroom management, three recommended readings [52–54] were provided for download on the university's content management system used for this course. As theory–practice integration was part of the treatment and a central dependent variable, we needed to control for different preconditions of theoretical knowledge caused by sources outside the treatment sessions. Students were thus asked which of the articles they had read. Assessing the literature read (and attendance as well as instructors' positive attitude toward the treatment, see below) based on self-report carries the risk of social desirability. To counteract this bias, we repeatedly emphasized the anonymity of the pretest and posttest. We further pointed out that the data will be used exclusively for research purposes and will not be shared with the instructors. The number of students who reported not having read any or only one of the texts can be interpreted as an indication of a minor role of social desirability: After the two treatment sessions, 41% of the students indicated they read none, and 28% read all three texts (median = 1). The theoretical literature read was used as a control variable.

**2.4.5 Attendance.**   As the treatment was part of two sessions of a regular course, student attendance varied and influenced the efficacy of the treatment. The more sessions they attended, the more the treatment could influence their cognition and behavior. Student attendance is therefore an aspect of treatment feasibility [58]. In a field-based setting, we cannot determine or standardize student attendance, consequently, we measured it. Accordingly, we asked the students how many of the two treatment sessions they attended (none, one, or two). The maximum of two sessions were attended by 79%, whereas 20% attended one session, and

1% attended neither of the two sessions. Again, these numbers are indicative (but no evidence) of low social desirability. Attendance was used as a control variable.

**2.4.6 Instructors' positive attitude toward the treatment.**   During the yearlong training instructors received for the intervention, we noticed their divergent attitudes toward the two instructional approaches. Attitudes largely reflect the extent to which the instructional approach is consistent with the values and practice routines of the instructors [59]. Particularly in field settings, measures of attitudes are associated with values and practice routines. Consequently, the interrelationships of these constructs must be taken into account when interpreting their results. To assess their attitudes, we separately asked for both treatments they conducted: "Think about the concept of the PB [DI] course: I like the way learning with classroom vignettes is handled here." Neither treatment tended to be more popular with all instructors: On the 6-point Likert scale (1 = disagree strongly to 6 = agree strongly), instructors' attitudes ranged from 2 to 6 for the problem-based treatment (M = 4.77; SD = 1.42) and 1 to 6 concerning the direct instructional treatment (M = 3.80; SD = 1.84). We ran a Bayes factor t-test for dependent samples, which showed considerable evidence for difference between the treatment groups ($BF_{10} = 1.424 \cdot 10^7$). This indicates that instructors had divergent but not necessarily one-sided attitudes throughout for each treatment, thus, we used the variable as a covariate of the treatment. We matched the attitude of the instructor toward a specific course to students' data from exactly that course. This way we were able to predict student level data with the respective course level information.

## 2.5 Statistical analysis

The collected data contained 18% missing values overall. Therefore, in the first step, we decided to impute the data via chained equations with the R package 'mice' [60]. Second, with the complete datasets, we computed separate full structural equation models for each of the three dependent variables (selective attention, reflective thought, and theory–practice integration) with the predictors' treatment (0: DI, 1: PB), attendance, theoretical literature on classroom management, and instructors' attitudes about the treatment (Fig 2). Note that the three vignettes we used to measure the dependent variables in the pretest and posttest were matched with great effort, yet they were not exactly the same. As a result, comparisons between the pretest and posttest on absolute scores must be interpreted carefully for all three dependent variables. Hence, we preferred to use the pretest scores as predictors of the posttest, accounting for differences before the treatment. Other predictors can be interpreted as increasing or decreasing the posttest's selective attention score under the control of pretest scores. Reflecting our study design, we obtained clustered data. Given our research interest, we are not interested in modeling these clustered data, but consider them a nuisance [61]. Accordingly, we used robust standard errors and adjusted $\chi^2$ that take the clustered structure into account.

Fit indices of the three models were good: Taking the N = 638 students into account, it is not surprising that the $\chi^2$-test shows a significant result ($\chi^2_{(40)} = 56.743\text{–}67.988$, p = .008–.053). Furthermore, the CFI = .962–.981, TLI = .950–975, and RMSEA = .035–.033 with $CI_{95\%}$ = [.014–.042] indicated good fit (lowest and highest value respectively).

As mentioned in the hypothesis section, a Bayesian approach is needed to test the formulated hypotheses. Accordingly, with the results from the models, we applied a Bayesian informative hypothesis approach using the R package 'bain' [62]. As opposed to the commonly used frequentist null hypothesis significance testing, this allows us to quantify and compare relative evidence of hypotheses (including a null hypothesis). The hypotheses to be compared were derived from the formulated hypotheses above. We tested the hypotheses of the two predictors (instructional approaches, instructors' attitudes) within each dependent variable

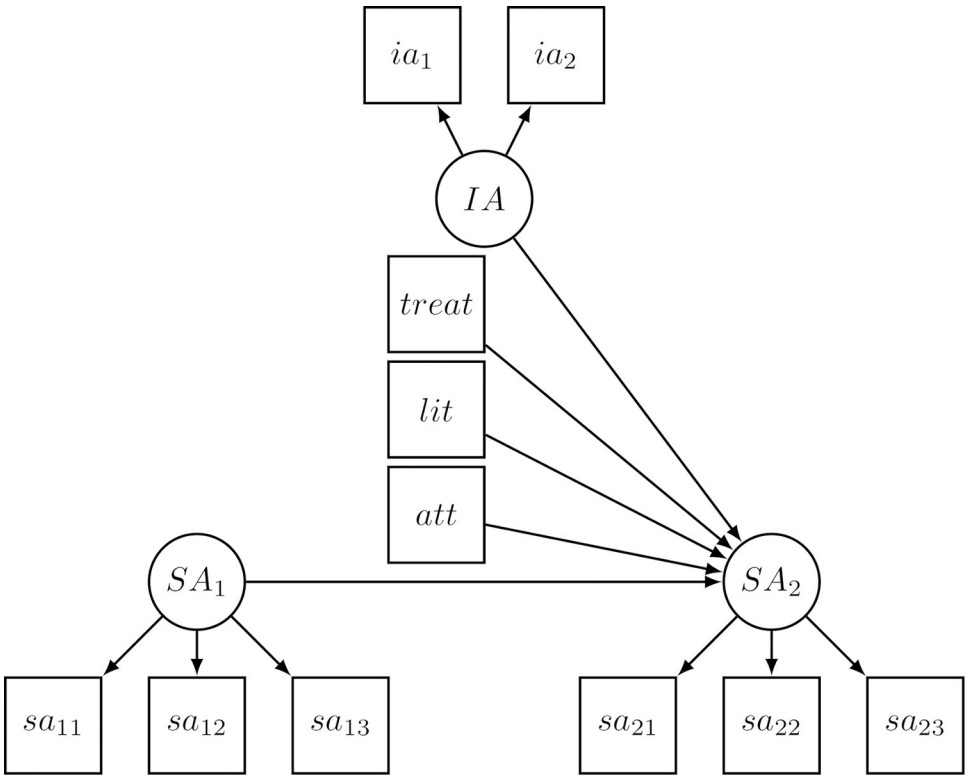

**Fig 2. Computed structure model for one dependent variable (i.e., selective attention, models on reflective thought, and theory–practice integration structured accordingly).** *SA*: Latent variable "selective attention" in pretest (*SA₁*) as a control variable and posttest (*SA₂*) as the dependent variable; *sa*: Manifest variables of "selective attention" representing the three vignettes within each measurement point; *att*: Attendance; *lit*: Theoretical literature on classroom management; *treat*: Treatment; *IA*: Latent variable of instructors' attitudes about the treatment; *ia*: Manifest variables of instructors' attitudes.

simultaneously, to increase rigor by making the predictions as precise as possible. They will be reported directly in the respective results part to reduce complexity (all analyses and results can be examined in the supplemental material).

## 3. Results

### 3.1 Selective attention

The number of selected situations slightly declined from pretest to posttest in both groups (Table 3). This might be an unexpected result; however, please be reminded that a direct comparison of pretest and posttest should be interpreted with caution.

We expected the DI and PB approaches to foster selective attention (no difference between groups) and for positive attitudes of the instructors about the instructional approach to be positively related to selective attention. Therefore, the statistical hypothesis to test can be formulated as $H1_1$: $\beta_{\text{treat}} = 0$ & $\beta_{\text{IA}} > 0$, with the dichotomous treatment variable coded as DI = 0 and PB = 1 (see Fig 2). This hypothesis will be tested against $H1_2$: $\beta_{\text{treat}} = 0$ & $\beta_{\text{IA}} < 0$ as one may also assume that instructors with positive attitudes toward a treatment made students analyze situations in greater detail. Students would, in consequence, have selected fewer situations to analyze. These hypotheses are further compared with a null hypothesis, $H1_0$: $\beta_{\text{treat}} = 0$ & $\beta_{\text{IA}} = 0$, and an unrestricted hypothesis that will have the greatest probability in case all our formulated hypotheses are implausible, $H1_u$: $\beta_{\text{treat}}$; $\beta_{\text{IA}}$.

**Table 3. Means (and standard deviations) of the three dependent variables.**

| | Selective attention | | Reflective thought | | Theory-practice integration | |
|---|---|---|---|---|---|---|
| | Pretest | Posttest | Pretest | Posttest | Pretest | Posttest |
| DI | 2.44 | 2.19 | 2.18 | 2.16 | .05 | .23 |
| | (1.71) | (1.41) | (.58) | (.59) | (.14) | (.33) |
| PB | 2.42 | 2.15 | 2.17 | 2.22 | .04 | .27 |
| | (1.81) | (1.44) | (.56) | (.60) | (.11) | (.34) |

*Note*: Selective attention: number of analyzed situations per test; reflective thought: realized inquiry steps in the analysis process [0–3]; theory–practice integration: relative frequency of analyses including theoretical target aspects

To describe the results, we indicate which hypothesis received the greatest posterior probability, then report the Bayes factor against its complement (opposite of the hypothesis) and the Bayes factors of the hypothesis against the other hypotheses that were also tested (see supplement for detailed results). Evidence pointed toward the exploratory hypothesis $H1_2$, as well as the null hypothesis $H1_0$. We found solid evidence of these hypotheses against their complement ($BF_{2c} = 27.32$; $BF_{0c} = 25.59$) and $H1_1$ ($BF_{21} = 50.97$; $BF_{01} = 47.73$). Comparing the two hypotheses $H1_2$ and $H1_0$ against each other yielded no clear result ($BF_{20} = 1.07$). We conclude that there is strong evidence that the two instructional approaches have an equivalent effect on selective attention. In addition, instructors' positive attitudes toward the treatment had either no relation or negative relation to the number of selected situations. However, regarding the instructors' attitude, we cannot make a concluding statement.

## 3.2 Reflective thought

Students' reflective thought (as measured by realized inquiry steps in the analyses) was already well developed before they entered the treatment sessions and showed only little change afterward.

We expected the PB approach to foster reflective thought (compared with DI) and instructors' positive attitude to show a positive relation, $H2_1$: $\beta_{\text{treat}} > 0$ & $\beta_{\text{IA}} > 0$. Two exploratory hypotheses tested whether only one of the effects holds, $H2_2$: $\beta_{\text{treat}} = 0$ & $\beta_{\text{IA}} > 0$ and $H2_3$: $\beta_{\text{treat}} > 0$ & $\beta_{\text{IA}} = 0$. As before, we included a null ($H2_0$) and an unrestricted hypothesis ($H2_u$).

As one may expect from the descriptive results, we found substantial evidence for the null hypothesis against the other hypotheses and its complement ($BF_{0u} = 82.00$; $BF_{0c} = 82.00$; $BF_{01} = 28.22$; $BF_{02} = 4.09$; $BF_{03} = 6.26$). Therefore, we conclude that the effect on students' reflective thought is equivalent between the instructional approaches, and the instructor's attitude is not related to students' reflective thought.

## 3.3 Theory–practice integration

Students' theory–practice integrations when analyzing classroom situations greatly changed from the pretest to the posttest ($\Delta M_{\text{DI}} = 18\%$; $\Delta M_{\text{PB}} = 23\%$). Even though the pretest and posttest vignettes are not identical, the difference in scores is certainly noteworthy. As opposed to reflective thought, students showed considerable room for potential with their theory–practice integrations in the pretest. A mere 2–8% of analyses on the vignettes in the pretest contained theory–practice integrations, although the theoretical literature on classroom management was already provided before the test. To obtain the effect of the treatment independently of the amount of literature read by the students, we measured and controlled for this variable (see Fig 2).

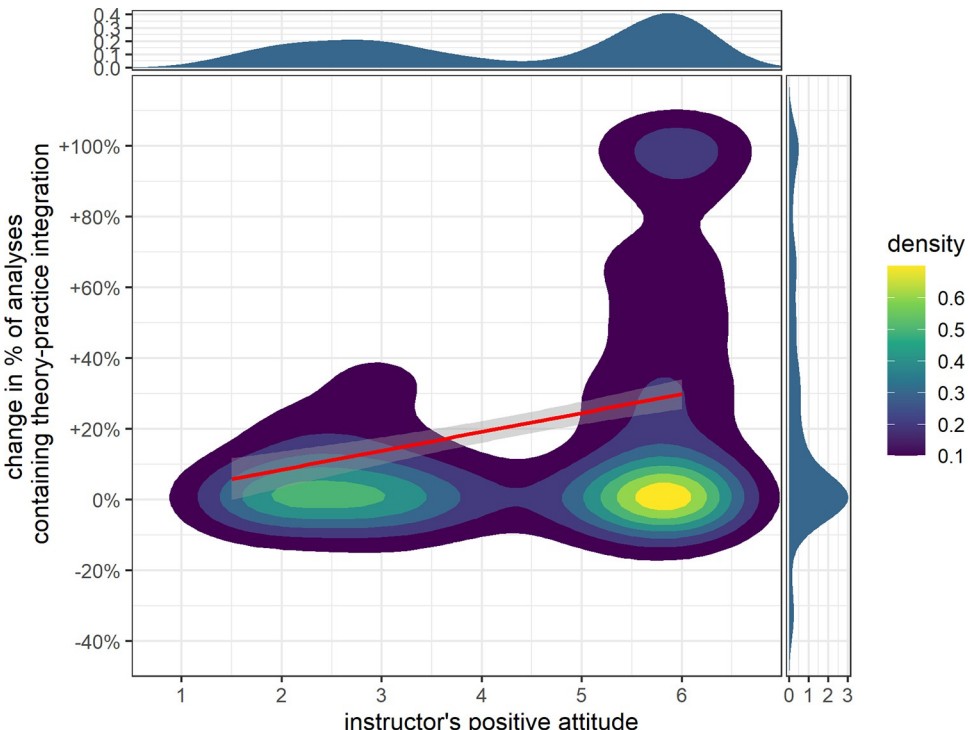

**Fig 3. Two-dimensional density plot of change from pretest to posttest in theory-practice integrations and the instructors' positive attitudes.**

We expected the PB approach to generate a similar amount of theory–practice integration as DI and the instructors' attitudes to be positively related, $H3_1$: $\beta_{\text{treat}} = 0$ & $\beta_{\text{IA}} > 0$. Like before, we explored further hypotheses on whether PB instruction shows a positive effect $H3_2$: $\beta_{\text{treat}} > 0$ & $\beta_{\text{IA}} > 0$ or the attitudes make no difference $H3_3$: $\beta_{\text{treat}} > 0$ & $\beta_{\text{IA}} = 0$, and included a null ($H3_0$) along with an unrestricted hypothesis ($H3_u$). Evidence indicates strong support for the hypothesis $H3_1$ ($BF_{1u} = 34.06$; $BF_{1c} = 34.06$; $BF_{12} = 12.24$; $BF_{13} = 90.95$; $BF_{10} = 41.09$). From these results, we infer that both instructional approaches lead to an equivalent effect on students' theory–practice integrations. What is more, instructors' positive attitudes toward the treatment are related to an increase in theory–practice integrations (see Fig 3).

## 4. Discussion

### 4.1 Interpretation of results

The goal of our field study was to compare different learning scenarios using authentic representations of practice and their differential effects on how students analyze classroom situations. More specifically, we aimed to consider selective attention, reflective thought, and theory–practice integration. Our data did not reveal differences between the DI and PB approaches on the selective attention of students. At first glance, the treatment's low impact does not necessarily contradict Seidel et al.'s [42] results because students in the DI courses might have been able to notice more critical situations, but if given a choice, did not make use of that skill due to negative attitude or a lack of motivation. To test for this possible explanation, we included the students' willingness for effort and their attitude on readiness for reflection in the structure model. The variables had no significant relation with selective attention, and thus, cannot resolve this issue. Furthermore, we found that the number of selected situations decreased from the pretest

to the posttest in both conditions. We cannot conclusively elucidate this phenomenon with our data, but we offer some tentative interpretations. A first intuitive explanation is that the analyses became fewer because students wrote longer analyses. However, we did not observe an increase in the length of the analyses in our data. A second explanation could be that the vignettes to be analyzed in the pretest and the posttest offer different numbers of situations that can be analyzed. Although we matched the vignettes to the pretest and posttest with great effort, we cannot exclude this option. A third explanatory approach relates to test fatigue. The students may put more effort into the pretest because analyzing classroom videos was a novel format for them (novelty effect). After they went through the pretest and analyzed several instructional videos again in the treatment sessions, the novelty effect may have worn off and their willingness to reflect may have decreased in the posttest. A slight decrease in scales of readiness to reflect was indeed observed in our data. Concerning the relation of the instructors' attitude toward the instructional approaches with the selective attention of students, we were not able to make a final conclusion, as results from the data were inconclusive.

We were able to reveal evidence for PB and DI to have equivalent effects on the reflective thought of students. This contradicts our assumptions formulated before data collection. As the strongest predictor in the model was the pretest score on reflective thought and our intervention was rather short (two 90-minute sessions), this might support assumptions that reflective thought is rather a stable skill, and thus, more challenging to influence. Further inquiry shows a negative relationship with the number of selected situations ($r = -.371$, $p < .001$) and a positive correlation with theory–practice integration ($r = .278$, $p < .001$). The motto for high performers in inquiry seems to be less quantity, more quality. Overall, the means imply consistent high scores on the 0–3 scale from pre- to posttest for both groups. Moreover, instructors' attitudes toward the treatment did not play a role in students' reflective thought.

Students' theory-practice integration scores in analyzing classroom situations improved greatly from the pretest to the posttest. However, this result should be taken with a grain of salt because the pretest and posttest are not equivalent, even though we matched them with great effort. Both instructional approaches appear to have equivalent positive effects, as attendance in either represents a significant predictor. Thus, these results do not support Dochy et al.'s [40] and Seidel et al.'s [42] findings. Interestingly, the strongest predictor (stronger than reading literature on the topic) represents the instructors' attitude toward the instructional approaches. What is more, their attitudes were rather heterogeneous: The instructors favored different instructional approaches with no approach being universally preferred.

## 4.2 Limitations and further research

Regarding implications for practice, it is important to keep in mind that while we captured performance in the analysis of practice, we did not capture whether this analysis had an impact on student teachers' classroom practice. With the data and design of the current study, we cannot draw inferences on this relation. However, there is first empirical evidence that the development of analysis skills has a positive impact on classroom practice [63, 64].

Further, in our measurement tool, we operationalized selective attention as the number of comments students gave on the classroom vignettes. This conceptualization makes it difficult to compare the data with further studies in the field, such as those on professional vision [65]. With our performance-based operationalization, differences in the number of selected situations may be interpreted as differences in the ability to notice critical situations or as differences in motivation or attitude. We tried to address this limitation by including related variables (willingness for effort, readiness for reflection), but this did not improve the model or influence correlations between latent variables.

Our results underscore the significance of Baecher et al.'s [43] claim that more attention should be paid to the role of instructors. However, it remains unclear how the instructors' attitudes about learning scenarios affect the students' performance in applying the analysis of practice. Further insight and research are needed on the instructors' and students' sides to clarify the path of effects and interactions: How does the instructors' attitude influence their teaching performance and how does the teaching performance influence the students' beliefs and performances? Lastly, the treatment was short compared with, for example, video clubs [21]; this was due to the field study character of the teacher education program. An artificially prolonged treatment that exceeded the courses' two sessions on classroom management could have had a different effect, but this would have reduced the external validity in our case.

### 4.3 Conclusion

With the limitations in mind, we draw two major conclusions from our data. First, we refer to the question posed in the title: Do direct instructional and problem-based approaches really produce different effects? Based on our data, the answer is *no*. In our study, we produced evidence, that using either DI or PB approaches to short-term interventions will yield equivalent effects on students' selective attention, reflective thought, and theory–practice integrations. Particularly, students' reflective thought proved to be a stable skill, and therefore, would need to be addressed with longer interventions (e.g., over one semester).

Second, encouragingly, both instructional approaches can foster theory–practice integrations of students, with the instructors playing a key role. These results contribute to further uncovering approaches to increase theory–practice integration, which was already labeled a "highly relevant endeavor" [66]. They also underscore the importance of examining the role of instructors within future field-based research. Based on our findings, we might not necessarily recommend forcing instructors to use certain (allegedly effective) learning methods, but rather, to draw on those about which they have positive views. We consider this result as a vital insight because it should be relevant for related field studies or the interpretation of results in laboratory studies (e.g., where researchers function as instructors). Coming from a field study design perspective, these results are applicable for teacher education practice, and thus, highly relevant.

## Supporting information

**S1 File.**
(HTML)

## Author Contributions

**Conceptualization:** Jürgen Schneider, Marc Kleinknecht, Thorsten Bohl, Sebastian Kuntze, Markus Rehm, Marcus Syring.

**Data curation:** Jürgen Schneider.

**Formal analysis:** Jürgen Schneider.

**Funding acquisition:** Marc Kleinknecht, Thorsten Bohl, Sebastian Kuntze, Markus Rehm.

**Investigation:** Jürgen Schneider, Marcus Syring.

**Methodology:** Jürgen Schneider, Marcus Syring.

**Project administration:** Jürgen Schneider.

**Resources:** Marcus Syring.

**Software:** Jürgen Schneider.

**Supervision:** Marc Kleinknecht, Thorsten Bohl, Sebastian Kuntze, Markus Rehm.

**Visualization:** Jürgen Schneider.

**Writing – original draft:** Jürgen Schneider.

**Writing – review & editing:** Marc Kleinknecht, Thorsten Bohl, Sebastian Kuntze, Markus Rehm, Marcus Syring.

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
