## [Decision Letter · Decision Letter 0]

21 Dec 2021

PONE-D-21-30051Using authentic representations of practice in teacher education: Do direct instructional and problem-based approaches really produce different effects?PLOS ONE

Dear Dr. Schneider,

Thank you for submitting your manuscript to PLOS ONE. After careful consideration, we feel that it has merit but does not fully meet PLOS ONE’s publication criteria as it currently stands. Therefore, we invite you to submit a revised version of the manuscript that addresses the points raised during the review process.

I agree with the reviewers' thoughts that this was a well conducted study with a impressive sized sample, but that there are also several places to improve the work. Please carefully consider and address the two reviewer's comments below in your next submission, I do not repeat them here. I note that Reviewer 1 put their comments in an attached doc; very little is directly in the letter. In addition, I will point out that even though in your submission information you included a link to the data set, both reviewers indicated you did not include access to the raw data. So, I think you need to in the body of the manuscript, make it quite clear how to access the relevant data. 

We look forward to receiving your revised manuscript.

Kind regards,

Micah B. Goldwater, Ph.D

Academic Editor

PLOS ONE

Journal Requirements:

2. Please consider changing the title so as to meet our title format requirement (https://journals.plos.org/plosone/s/submission-guidelines). In particular, the title should be "Specific, descriptive, concise, and comprehensible to readers outside the field" and in this case it is not informative and specific about your study's scope and methodology

3. Please provide additional details regarding participant consent. In the ethics statement in the Methods and online submission information, please ensure that you have specified (1) whether consent was informed and (2) what type you obtained (for instance, written or verbal, and if verbal, how it was documented and witnessed). If the need for consent was waived by the ethics committee, please include this information

4. Peer review at PLOS ONE is not double-blinded (https://journals.plos.org/plosone/s/editorial-and-peer-review-process). For this reason, authors should include in the revised manuscript all the information removed for blind review.

Reviewers' comments:

Reviewer's Responses to Questions

**Comments to the Author**

1. Is the manuscript technically sound, and do the data support the conclusions?

Reviewer #1: Yes

Reviewer #2: No

2. Has the statistical analysis been performed appropriately and rigorously? 

Reviewer #1: Yes

Reviewer #2: Yes

3. Have the authors made all data underlying the findings in their manuscript fully available?

Reviewer #1: No

Reviewer #2: No

4. Is the manuscript presented in an intelligible fashion and written in standard English?

Reviewer #1: No

Reviewer #2: No

5. Review Comments to the Author

Reviewer #1: Thank you for letting me review this manuscript.

I congratulate you to a sound study with a large sample size. Yet I would see the manuscript to get developed further. It would improve greatly by using a more precise language.

I also tried to reproduce your results but I was missing the dataset "delete.Rdata". I liked that you provided a markdown file with all your code.

Many more comments attached :)

Greetings

Reviewer #2: The authors addressed an interesting topic of comparing students’ skill development of selective attention, reflective thought and theory-practice integration between two different learning approaches – Direct Instruction (DI) and Problem-Based Learning (PBL). I appreciate your effort in this study. However, there are a few areas that I would like the authors to consider before publishing the paper.

1. The authors may need to restructure the manuscript and add a literature review on PBL and DI in teacher education. For example, the definition of PBL. What does other research already know about PBL and DI? Why are the skills of selective attention, refective thought and theory-practice integration important to students in teacher education? Are they difficult to be developed? Why do the authors anticipate PBL or DI could help students develop those three skills. Moreover, some contents in the Method section should be moved to the literature review session. For instance, the first part of 3.4.1 (line 403 to 420) was the literature review about “Selective Attention”. Similarly, the first paragraph of 3.4.2, 3.4.3 and 3.4.4 should move to the literature review section.

2. There are no research questions, only hypotheses in this study. However, the authors might need to provide the relational or evidence of the predictions based on previous research.

3. The sentence in Line 314 is not clear.

4. LINE 281: What is the reason to redesign sessions 6 and 7, rather than other sessions?

5. Line 343: It is not clear that students analysed the situations in groups or individually.

6. In Line 352, the authors mentioned that “To guide the analysis, students received key questions that targeted the analysis of practice steps”. It means students were received the guidance of the analysis procedure, step by step. Based on one of the essential characteristics of PBL, “ The problem simulations used in problem-based learning must be ill-structured and allow for free inquiry (p.13, Savery, J.R., 2006)”. It seems the treatment in the PBL group was not ill-structured and didn’t allow for free inquiry.

Savery, J. R. (2015). Overview of problem-based learning: Definitions and distinctions. Essential readings in problem-based learning: Exploring and extending the legacy of Howard S. Barrows, 9(2), 5-15.

7. Line 379: Not sure where are the research questions of this study?

8. In section 4.1, I am keen to know why the number of analysed situations decreased in both groups.

9. In section 4.2, the authors mentioned that “Students’ reflective thought (as measured by realized inquiry steps in the analyses) was already well developed before they entered the treatment sessions”. If this is the case, why did the authors measure their skills of reflective thoughts? The authors already know students have this skill before the intervention.

10. Line 607: The authors mentioned that “ we conclude that the effect on students’ reflective thought is equivalent between learning approaches”. The authors had explained because students had already developed the skill of reflective thoughts before the interventions; therefore, there was no significant difference between the pre-and post-tests. Then the authors conclude that the effect on students’ reflective thoughts is equivalent between learning approaches. Please indicate what evidence to make this conclusion.

11. There is no discussion in the Discussion section.

I hope the authors find these comments useful.

6. PLOS authors have the option to publish the peer review history of their article (what does this mean?). If published, this will include your full peer review and any attached files.

Reviewer #1: No

Reviewer #2: No

---

## [Author Response · Author response to Decision Letter 0]

30 May 2022

Reviewer #1

I also tried to reproduce your results but I was missing the data set "delete.Rdata". I liked that you provided a markdown file with all your code.

Our Response

Thank you for bringing this problem to our attention. We unnecessarily blinded the document with the analyses. The link to both data sets is now in the RMarkdown file. We also put the "delete.RDdta" data set on github in the "data_public" folder and renamed it to "ts.Rdata". How to download the data can be found under the heading "Import data" in the RMarkdown file.

I think the summary of the reviews could be more concise, more to the point. There is no advantage of citing that researchers are "reinventing the wheel" (ll125f).

Our Response

We agree that the literature review would benefit from being restructured and rewritten for clarity. Therefore, we restructured and rewrote the introductory section, synthesizing the results of the reviews and tailoring literature review to the research questions.

We have deleted the statement on researchers "reinventing the wheel".

You claim to compare Problem Based Learning and Direct Instruction and your description of the methods you used fits these terms. However, in my particular understanding the definition of DI from Kirschner (2006) that you bring, would only define a “darstellende Stoffvermittlung”. In the current agreement of educational science, Direct Instruction names a specific instruction method that includes multiple phases.

Our Response

We agree that the definition by Kirschner (2006) is too narrow for our conceptualization of direct instruction. We changed this statement to:

“In contrast, DI describes a teacher-centered approach in which phases of modeling are typically followed by phases of guided and individual practice (Stockard et al., 2018).” The hypotheses are clear and could be tackled with this data. It would help the understanding of the results, if you already here make clear that you always test the sub-hypotheses in combination (and give a reason for it).

Our Response

We now included a statement below the description of hypotheses: “We tested the hypotheses of the two predictors (instructional approaches, instructors’ attitudes) within each dependent variable simultaneously to increase rigor by making the predictions as precise as possible.”

You use the number of attended sessions, the number of articles read for these sessions and instructor’s attitude towards the instruction method as proxies/control variables. However, it remains unclear, what these proxies are really measuring. A more thorough discussion would be indicated.

Our Response

We agree that a more thorough discussion of these measures would be indicated. Therefore, we added paragraphs on the social desirability and interpretation of these measures in the sections 2.4.4, 2.4.5 and 2.4.6.

From a pedagogical point of view I do not really find the task described in lines 329-331 as promoting learning. A good task (Arbeitsauftrag) should be structured more and be more elaborate.

Our Response

We explicitly refrained from strongly structuring the task, especially against the background of the PB instruction. The instructors implemented the task in the sense of their instructional approach.

I wondered whether each instructor had courses in both conditions. Please specify.

Our Response

We agree that this information was missing in the manuscript. We now added a sentence to the “Design” section:

“Each instructor taught DI an PB courses, the conditions were balanced within the instructors (teaching the same amount of both conditions, except when teaching an odd number of courses).” In section 3.4.1. there appeared again a short literature overview. I would shift this to the introduction above.

Our Response

We agree, thank you for bringing this to our attention. These contents have now been moved to the literature review section and integrated accordingly.

In section 3.4.2 I wondered why you tested for unidimensionality of the vignettes? Do you mean the raters score of reflective thought per vignette? Why did you only test for unidimensionality of reflective thought and not of selective attention as well?

Our Response

We clarified this section by changing one sentence to “The raters’ scores were tested for one-dimensionality per vignette”. When coding qualitative data, it is useful to check for dimensionality. One-dimensionality may indicate that the coding does not covary with other factors (e.g., the part of the vignette to which the analysis being coded refers). Selective attention was not tested for one-dimensionality as the score comprises the sum (count) of analyses regarding classroom management. Therefore, a calculation of the dimensionality is not possible. Theory–practice integration was not tested for one-dimensionality for the same reason.

When you report interrater reliability, please also specify how many cases have been rated by two (or even three) raters. Please also state what you did in cases in which the raters had different scores.

Our Response

We agree that this is essential information for assessing interrater reliability. We have updated the sentence accordingly: 

“Inter-rater reliabilities for all codings were computed based on randomly selected 20% of the approximately 7 600 comments written by the participants in the pretest and posttest. Cohen’s Kappa scores of the two trained raters were satisfactory (κ = .64–.77) and disagreements were resolved through discussion.” When reading the sentence in ll. 395 it was unclear to me what you meant by it. After reading part 3.5, I understood that you aimed at matching similar vignettes to pre- and posttest. Please clarify.

Our Response

We concede that the description might have been unclear. We restructured this section and added the sentence: “We investigated the extent to which pairs of similar vignettes could be found from a classroom video, each of which was then split between the pretest and the posttest.”

Please make clearer, that you test the two sub-hypotheses of each dependent variable in combination. Currently, the indices of the hypotheses do not match and it is difficult to match the hypotheses and results. I would number the three main hypotheses with numbers from 1-3 and the different cases that you test (hypothesized direction, opposite direction, null hypothesis, unrestricted hypothesis) with letters from a-d for example.

Our Response

We agree that the consistency can be increased by numbering the hypotheses and sub-hypotheses. Numbers and indices of hypotheses in the section “Research Questions and Hypotheses” now match those from the “Results” section. Further, we numbered hypotheses on selective attention as H1 (H11, H12, H10, H1u), hypotheses on reflective thought as H2 (H21, H22, H23, H20, H2u) and hypotheses on theory-practice integration as H3 (H31, H32, H33, H30, H3u). The hypotheses we formulated in the section " Research Questions and Hypotheses" are each assigned the index 1 (H11, H21, H31). Also, we now mention that the sub-hypotheses are tested simultaneously in the sections “Research Questions and Hypotheses” as well as in “Statistical Analyses”.

Figure 2: Would it be possible to depict the raw data instead of the 12 data points here? It is not really interesting how the theory-practice integration differs by the vignettes, but more how it differs by DI vs PBL and how strongly students vary in that. Raw data per student would be very interesting.

Our Response

We assume you refer to Fig. 3: A similar graph with raw data would be an interesting option, but is unfortunately very cluttered. Therefore, we calculated the change scores for each person on the two variables and created a two-dimensional density plot of change, differentiated for treatment groups. We updated the figure and caption accordingly.

You state that you included the students’ willingness for effort and their attitude on readiness for reflection, but I did not see any results regarding this. Is this reported in the supplementary material.

Our Response

In the supplementary material we only included analyses directly mirrored in the manuscript. The inclusion of further exploratory analyses would overload the document.

You state “Students’ theory–practice integrations when analyzing classroom situations greatly increased from pretest to posttest.” Beforehand you ask the reader to be careful when comparing the scores between pre- and posttest, as the vignettes differ. I thus find that this conclusion is a bit far fetched. It could just be due to the different vignettes.

Our Response

We agree that this result should be interpreted with greater caution. We rewrote the sentence as follows: “Students' theory-practice integration scores in analyzing classroom situations improved greatly from the pretest to the posttest. However, this result should be taken with a grain of salt because the pretest and posttest are not equivalent, even though we matched them with great effort.”

I very much liked that the supplementary material contains a knitted R markdown file containing all the code and results. I wondered why the authors put the data on the gesis server in a proprietary .sav format (maybe add a .txt or .csv file as well). Please also upload the data “delete.Rdata”, as this is vital to reproduce your code.

Our Response

We agree that a proprietary file format (such as .sav) is less accessible than open file formats (such as .Rdata). At the recommendation of GESIS, we uploaded a SAV file (and not a CSV-file) to the repository. As opposed to CSV, SAV-files include the item labels and levels – but so does the .Rdata file format. This is why we also uploaded the data sets as .Rdata. The "delete.RDdta" data set is now on github in the "data_public" folder and we renamed it to "ts.Rdata". A guide on how to download the data can be found under the heading "Import data" in the RMarkdown file. The links to both relevant data sets are now in the RMarkdown file.

I suggest a sound proofreading focusing of the clarity and precision of the language. Also please make use of the past tense, and use it consistently. You may also consider asking the writing experts at your institution. From my experience, this is always greatly improving a manuscript.

Our Response

We have thoroughly proofread the manuscript and revised it accordingly. For this we also involved an external writing expert.

Use key nouns consistently: for example, the manuscript uses “DI and PB methods”/“PB instruction”/“PB learning”/“PBL” → choose one. This also happens for “subject”/“student”/“teacher student” and “count”/“number”/“quantity” and “second semester”/“2nd semester”/“teachers grade level (second)”. Please be very precise and consistent in the words you use.

Our Response

We agree that consistency will foster clarity of the manuscript. We decided for “DI/PB approach”, “student” (however, we kept “student teachers” when it contributed to understanding), “number of selected situations”, “second semester” and updated the manuscript accordingly.

I would not number the overview of current research as “2.”, but put it as sub-sections of the Introduction.

Our Response

We updated the manuscript accordingly.

Reviewer #2

The authors may need to restructure the manuscript and add a literature review on PBL and DI in teacher education. For example, the definition of PBL. What does other research already know about PBL and DI? Why are the skills of selective attention, reflective thought and theory-practice integration important to students in teacher education? Are they difficult to be developed? Why do the authors anticipate PBL or DI could help students develop those three skills.

Our Response

We have rewritten the introductory section and added remarks on the relevance of reflection and related constructs in teacher education. For each section of selective attention, reflective thinking, and integration of theory and practice we elaborated their relevance to teacher professionalism and the development of professionalism (section “Selective attention, reflective thought and theory-practice integration with authentic representations of practice”). Further you will now find theoretical and empirical elaborations on the efficacy of PBL and DI to develop these skills in the “Problem-based and direct instruction” section.

Moreover, some contents in the Method section should be moved to the literature review session. For instance, the first part of 3.4.1 (line 403 to 420) was the literature review about “Selective Attention”. Similarly, the first paragraph of 3.4.2, 3.4.3 and 3.4.4 should move to the literature review section.

Our Response

We agree, thank you for bringing this to our attention. These contents have now been moved to the literature review section and integrated accordingly.

There are no research questions, only hypotheses in this study. However, the authors might need to provide the relational or evidence of the predictions based on previous research.

Our Response

We now included research questions as well as hypotheses. Also, we have updated the introductory section to include literature that leads more directly to the predictions of the hypotheses.

The sentence in Line 314 is not clear.

Our Response

We deleted the sentence “In the two sessions, students learned about the classroom management strategies of Kounin (1970), Evertson (2006), and Mayr (2009).” and described the contents of the course in the design section.

LINE 281: What is the reason to redesign sessions 6 and 7, rather than other sessions?

Our Response

We focused on redesigning sessions where classroom management was on the curriculum. This was the case for sessions six and seven. We updated the sentences “For the interventions, we redesigned two of the courses’ weekly 90-minute sessions (6th and 7th of 15 sessions) and an assignment between these two sessions using authentic representations of practice. The topic of these two sessions and the assignment was classroom management.” to “For the interventions, we redesigned part of the courses (two sessions and an inter-session assignment) using authentic representations of practice. For this purpose, we focused on sessions in which classroom management was on the curriculum. These were sessions six and seven of 15 sessions.”

Line 343: It is not clear that students analysed the situations in groups or individually.

Our Response

Thank you for bringing this to our attention. We changed the sentence to “After this, students individually analyzed some more situations”

In Line 352, the authors mentioned that “To guide the analysis, students received key questions that targeted the analysis of practice steps”. It means students were received the guidance of the analysis procedure, step by step. Based on one of the essential characteristics of PBL, “ The problem simulations used in problem-based learning must be ill-structured and allow for free inquiry (p.13, Savery, J.R., 2006)”. It seems the treatment in the PBL group was not ill-structured and didn’t allow for free inquiry.

Our Response

Our description in the manuscript may have been somewhat unclear: The problem is ill-structured as students determine what they consider to be a problem (selection of situations) and what is and is not part of the problem. The students were free to inquire solutions to these situations. The key questions merely served as a guide in case students needed support with their inquiry. The questions did not serve as step-by-step instructions and were not introduced as such. We now clarified this in the manuscript.

Line 379: Not sure where are the research questions of this study?

Our Response

We now included research questions.

In section 4.1, I am keen to know why the number of analysed situations decreased in both groups.

Our Response

This is indeed an interesting phenomenon that needs further investigation. However, our data unfortunately do not allow us to answer this question conclusively. We now offer several interpretations of this in the discussion section: „Furthermore, we found that the number of selected situations decreased from the pretest to the posttest in both conditions. We cannot conclusively elucidate this phenomenon with our data, but we offer some tentative interpretations. A first intuitive explanation is that the analyses became fewer because students wrote longer analyses. However, we did not observe an increase in the length of the analyses in our data. A second explanation could be that the vignettes to be analyzed in the pretest and the posttest offer different numbers of situations that can be analyzed. Although we matched the vignettes to the pretest and posttest with great effort, we cannot exclude this option. A third explanatory approach relates to test fatigue. The students may put more effort into the pretest because analyzing classroom videos was a novel format for them (novelty effect). After they went through the pretest and analyzed several instructional videos again in the treatment sessions, the novelty effect may have worn off and their willingness to reflect may have decreased in the posttest. A slight decrease in scales of readiness to reflect was indeed observed in our data. “

In section 4.2, the authors mentioned that “Students’ reflective thought (as measured by realized inquiry steps in the analyses) was already well developed before they entered the treatment sessions”. If this is the case, why did the authors measure their skills of reflective thoughts? The authors already know students have this skill before the intervention.

Our Response

The time period that the students were given to complete the pretest extended until directly before the first treatment session. Accordingly, we were unfortunately not able to analyze the data before the treatment.

Line 607: The authors mentioned that “ we conclude that the effect on students’ reflective thought is equivalent between learning approaches”. The authors had explained because students had already developed the skill of reflective thoughts before the interventions; therefore, there was no significant difference between the pre-and post-tests. Then the authors conclude that the effect on students’ reflective thoughts is equivalent between learning approaches. Please indicate what evidence to make this conclusion.

Our Response

The conclusion “that the effect on students’ reflective thought is equivalent between learning approaches” is not related to our assumption that students had already developed the skill of reflective thoughts before the interventions. We derived this conclusion directly from our data: The inferential statistical comparison of the formulated hypotheses generated evidence for the null hypothesis. The null hypothesis states that there is no difference between the two groups. We therefore generated evidence that the effect is equivalent, which is possible with Bayes Factor hypothesis testing.

There is no discussion in the Discussion section.

Our Response

We added a discussion on the decrease of the number of selected situations. Further, we added a paragraph on the relation of the analysis of practice and classroom practice itself.

---

## [Decision Letter · Decision Letter 1]

6 Jul 2022

PONE-D-21-30051R1Using authentic representations of practice in teacher education: Do direct instructional and problem-based approaches really produce different effects?PLOS ONE

Dear Dr. Schneider,

Thank you for submitting your manuscript to PLOS ONE. I and the reviewer agree that your have improved the manuscript from its first submission, but still  does not fully meet PLOS ONE’s publication criteria as it currently stands. Therefore, we invite you to submit a revised version of the manuscript that addresses the points raised during the review process. The reviewer has gone above and beyond what reviewers typically do in how they carefully considered your analyses and manuscript. I hope you are as grateful as I am for such an effort to improve your paper. Please carefully address each of their suggestions in your revision, if you decide to submit one. 

We look forward to receiving your revised manuscript.

Kind regards,

Micah B. Goldwater, Ph.D

Academic Editor

PLOS ONE

Reviewers' comments:

Reviewer's Responses to Questions

**Comments to the Author**

1. If the authors have adequately addressed your comments raised in a previous round of review and you feel that this manuscript is now acceptable for publication, you may indicate that here to bypass the “Comments to the Author” section, enter your conflict of interest statement in the “Confidential to Editor” section, and submit your "Accept" recommendation.

Reviewer #1: (No Response)

2. Is the manuscript technically sound, and do the data support the conclusions?

Reviewer #1: Partly

3. Has the statistical analysis been performed appropriately and rigorously? 

Reviewer #1: No

4. Have the authors made all data underlying the findings in their manuscript fully available?

Reviewer #1: Yes

5. Is the manuscript presented in an intelligible fashion and written in standard English?

Reviewer #1: Yes

6. Review Comments to the Author

Reviewer #1: Hey,

I saw that you integrated all of the comments and added essential information to the manuscript. It has become much better now :)

Thanks also for providing the necessary data to reproduce the analyses. I really appreciate that you make your code and data openly accessible; that's a great example of open science, which I also promote myself!

I could replicate your results in terms of BayesFactors. I did not reproduce your core results, as the multiple imputation did not end in a reasonable amount of time on my computer ;)

But I had a look at the variables with which you constructed the instructor's attitudes, and here I see the major problem:

First, the variable labels are unclear (doz_gef & doz_pass), and even lead to the suspicion, that these variables did not measure the attitude towards problem-based learning and direct instruction. Why did you choose these labels and which instructional approach do they refer to?

Secondly, the dataset "rating-treatmentcheck.sav" contains the real names of the instructors  anonymise and give them a code or number.

Thirdly, and this is the major problem: the instructors were not consistent in their ratings about their attitudes. For example, the instructor of seminars 10, 11, and 12 gave the following answers on the attitudes: 3-4, 1-2, 4-4.

Thus, depending on which group s/he taught, s/he had different attitudes toward the instructional approach!

This has many theoretical & practical implications for your research:

-the attitudes were not reliably measured; they varied, based on the seminar that was taught

-If the seminar groups vary in size, this might produce a bias in the multiple imputations (the answers from larger seminars will likely be weighed more?)

-If instructors themselves were not sure about their attitude, what does it mean for the whole study?!

I hope I did not misunderstand something here and am on the wrong track. If so, let me know. But I think you should delve into this issue and compare the instructors' attitudes across treatments. Maybe you can come up with an average per instructor across seminars, but it will be needed to be discussed thoroughly.

Now to the manuscript...

Importantly, the line numbers refer to the document with track changes!

line 698 ff. That is related to the instructors' attitudes again: I am not convinced by the argument that the broad range of attitudes indicates the absence of any bias of the instructors.

Please state the mean attitude per treatment & SDs, and maybe consider a Bayesian t-test for dependent samples to check, whether the attitudes are comparable (after you resolved the issue that they vary across seminars).

I am also not convinced to model these two distinct attitudes as one latent variable. It's not a truly reflective construct, is it? Wouldn't it be much more meaningful to see how attitudes towards DI predict theory-practice integration in the DI, whereas attitudes towards PB predict theory-practice integration in PB-approaches? Or maybe a difference score instead of a sum score? Not yet sure, but please explain in further detail, why you modeled the two attitudes as indicators of one underlying general attitude variable.

A second major aspect concerns Figure 3. To me, Fig 3 makes no sense; in the hypothesis you do not test whether theory-practice integration relate to realized inquiry steps, so why plot it? For which research question does it depict meaningful information? I would put instructors' attitude on the x-axis and then add a Figure 4 to plot the change in theory-practice integration (again predicted by instructors' attitude for example).

Minor aspects and typos:

l. 262: remove full stop before reference

l. 268: the reference seems to be in a smaller font size

ll. 365: As not all hypotheses are exploratory, I would start like that: "One of the hypotheses on reflective thinking (H21) was based on strong assumptions derived from theory. The rest of the hypotheses were labelled as exploratory, since robust research is lacking."

l. 426: according to APA, a sentence cannot start with a number in numeric form. You could circumvent this by saying "In total, 638..."

l. 437: "The study was conducted within our institute's teacher education program" -> you are from 4 different institutes :) I think it does not need to be stated that it was from any institute; just that it was in a regular teacher education program in Germany

l. 451: "(8am to 8pm)" (delete space)

ll. 451: "Each instructor taught DI and PB courses, the conditions" ('d' missing, use semicolon to separate these sentences)

ll. 461: As no 'standard' for priors exist, please replace "standard priors" by "default priors of the BayesFactor package (REF)"

l. 538: Bayes Factors of .5 are very weak/anecdotal evidence. Please put that into perspective

l. 662: give references to Kounin, Evertson, and Mayr or leave out

ll. 859: you state a substantial correlation of reflective thought and theory-practice integration. But did you report this correlation somewhere?? In Figure 3 there seems not any relation

ll. 886: if your operationalization of selective attention makes it difficult to compare to other literature (as you just counted the comments), then why did you choose it? What could future research improve??

In general, I found it a bit funny that it's single blind peer review (I can see your names), but the

references have been blinded. I guess that's unnecessary

I know that I expect a lot again, and I am fine if the authors provide good reasons why some things cannot be changed.

I hope my comments contribute to improve this work.

Best regards

7. PLOS authors have the option to publish the peer review history of their article (what does this mean?). If published, this will include your full peer review and any attached files.

Reviewer #1: **Yes: **Christian M. Thurn

---

## [Author Response · Author response to Decision Letter 1]

19 Jul 2022

Reviewer comment:

First, the variable labels are unclear (doz_gef & doz_pass), and even lead to the suspicion, that these variables did not measure the attitude towards problem-based learning and direct instruction. Why did you choose these labels and which instructional approach do they refer to?

Our response:

“doz” is short for “Dozierende” which signifies “instructor” in German. As the data set consists mostly of data from students we wanted to make this clear in the variable label. The suffix “_gef” and “_pass” refers to the two items measuring their attitudes perceived “pleasure” and perceived “fit”.

Reviewer comment:

Secondly, the dataset "rating-treatmentcheck.sav" contains the real names of the instructors  anonymise and give them a code or number.

Our response:

Thank you for this very important information. We followed your suggested procedure.

Reviewer comment:

Thirdly, and this is the major problem: the instructors were not consistent in their ratings about their attitudes. For example, the instructor of seminars 10, 11, and 12 gave the following answers on the attitudes: 3-4, 1-2, 4-4.

Our response:

I see how disagreement within instructors may seem like a substantial problem. However, the example you pointed out, come from within one instructor but between different treatments. Hence we would assume agreement (=similar ratings) within the same type of courses (=same treatment) but not necessarily between courses from different treatments within each instructor. This was the case here and is the case for all instructors. In the word document “Response to Reviewers” I have attached two figures showing the agreement/disagreement for the two items.

 

Reviewer comment:

line 698 ff. That is related to the instructors' attitudes again: I am not convinced by the argument that the broad range of attitudes indicates the absence of any bias of the instructors.

Our response:

Our wording “bias” might be somewhat unclear. We wanted to illustrate that attitudes varied considerably within the treatments signifying that each treatment did not experience “only disagree” or “only agree” and therefore instructors being one-sided throughout. We therefore changed the sentence to “This indicates that instructors had divergent but not necessarily one-sided attitudes throughout for each treatment, thus, we used the variable as a covariate of the treatment.”

Reviewer comment:

Please state the mean attitude per treatment & SDs, and maybe consider a Bayesian t-test for dependent samples to check, whether the attitudes are comparable (after you resolved the issue that they vary across seminars).

Our response:

We did not expect instructors to be similar in their attitudes toward the treatment. For this reason, we included the variable as a predictor. The decision to be made was solely between using the variable as a moderator or as a covariate. We added the Bayes factor test and deleted the part on the mediation as it does not add to the manuscript and might be unclear.

Reviewer comment:

I am also not convinced to model these two distinct attitudes as one latent variable. It's not a truly reflective construct, is it? Wouldn't it be much more meaningful to see how attitudes towards DI predict theory-practice integration in the DI, whereas attitudes towards PB predict theory-practice integration in PB-approaches? Or maybe a difference score instead of a sum score? Not yet sure, but please explain in further detail, why you modeled the two attitudes as indicators of one underlying general attitude variable.

Our response:

Our description might have been somewhat unclear. First, the attitude toward the treatment was measured via two items. These two items both measure how positive the attitude of the instructors was toward a treatment. As instructors taught several courses, they indicated their attitude toward the treatment for each of the courses separately. In the data set then, the attitude of the instructor toward a specific course was matched to students’ data from exactly that course. This way, we achieved exactly what you were suggesting: “attitudes towards DI predict theory-practice integration in the DI, whereas attitudes towards PB predict theory-practice integration in PB-approaches”. To clarify this, we added the following paragraph: “We matched the attitude of the instructor toward a specific course to students’ data from exactly that course. This way we were able to predict student level data with the respective course level information.”

Reviewer comment:

A second major aspect concerns Figure 3. To me, Fig 3 makes no sense; in the hypothesis you do not test whether theory-practice integration relate to realized inquiry steps, so why plot it? For which research question does it depict meaningful information? I would put instructors' attitude on the x-axis and then add a Figure 4 to plot the change in theory-practice integration (again predicted by instructors' attitude for example).

Our response:

We agree that the figure does not directly fit any of the hypotheses. We have therefore implemented the suggestion and plotted the change in theory-practice integration on the y-axis and the attitudes of the lecturers on the x-axis. We used a 2D density plot because scatter or point plots would be too cluttered. Further, we added density distributions of each of the two variables.

Minor aspects and typos

We corrected all minor aspects and typos according to your suggestions.

---

## [Decision Letter · Decision Letter 2]

22 Aug 2022

Using authentic representations of practice in teacher education: Do direct instructional and problem-based approaches really produce different effects?

PONE-D-21-30051R2

Dear Dr. Schneider,

We’re pleased to inform you that your manuscript has been judged scientifically suitable for publication and will be formally accepted for publication once it meets all outstanding technical requirements.

Kind regards,

Micah B. Goldwater, Ph.D

Academic Editor

PLOS ONE

Reviewers' comments:

Reviewer's Responses to Questions

**Comments to the Author**

1. If the authors have adequately addressed your comments raised in a previous round of review and you feel that this manuscript is now acceptable for publication, you may indicate that here to bypass the “Comments to the Author” section, enter your conflict of interest statement in the “Confidential to Editor” section, and submit your "Accept" recommendation.

Reviewer #1: All comments have been addressed

2. Is the manuscript technically sound, and do the data support the conclusions?

Reviewer #1: Yes

3. Has the statistical analysis been performed appropriately and rigorously? 

Reviewer #1: Yes

4. Have the authors made all data underlying the findings in their manuscript fully available?

Reviewer #1: Yes

5. Is the manuscript presented in an intelligible fashion and written in standard English?

Reviewer #1: Yes

6. Review Comments to the Author

Reviewer #1: (No Response)

7. PLOS authors have the option to publish the peer review history of their article (what does this mean?). If published, this will include your full peer review and any attached files.

Reviewer #1: **Yes: **Christian M. Thurn

---

## [Editor Report · Acceptance letter]

26 Aug 2022

PONE-D-21-30051R2 

Using authentic representations of practice in teacher education: Do direct instructional and problem-based approaches really produce different effects? 

Dear Dr. Schneider:

I'm pleased to inform you that your manuscript has been deemed suitable for publication in PLOS ONE. Congratulations! Your manuscript is now with our production department. 

Kind regards, 

on behalf of

Dr. Micah B. Goldwater 

Academic Editor

PLOS ONE